# Uncovering the Mechanisms of Active Components from Toad Venom against Hepatocellular Carcinoma Using Untargeted Metabolomics

**DOI:** 10.3390/molecules27227758

**Published:** 2022-11-10

**Authors:** Pan Liang, Yining Ma, Luyin Yang, Linshen Mao, Qin Sun, Changzhen Sun, Zengjin Liu, Maryam Mazhar, Sijin Yang, Wei Ren

**Affiliations:** 1National Traditional Chinese Medicine Clinical Research Base, Drug Research Center of Integrated Traditional Chinese and Western Medicine, The Affiliated Traditional Chinese Medicine Hospital of Southwest Medical University, Luzhou 646000, China; 2State Key Laboratories for Quality Research in Chinese Medicines, Faculty of Chinese Medicine, Macau University of Science and Technology, Macau 853, China; 3Institute of Integrated Chinese and Western Medicine, Southwest Medical University, Luzhou 646000, China

**Keywords:** toad venom, hepatocellular carcinoma, metabolomic, UHPLC-MS/MS, bufalin, cinobufagin

## Abstract

Toad venom, a dried product of secretion from *Bufo bufo gargarizans* Cantor or *Bufo melanostictus* Schneider, has had the therapeutic effects of hepatocellular carcinoma confirmed. Bufalin and cinobufagin were considered as the two most representative antitumor active components in toad venom. However, the underlying mechanisms of this antitumor effect have not been fully implemented, especially the changes in endogenous small molecules after treatment. Therefore, this study was designed to explore the intrinsic mechanism on hepatocellular carcinoma after the cotreatment of bufalin and cinobufagin based on untargeted tumor metabolomics. Ultraperformance liquid chromatography with tandem mass spectrometry (UHPLC-MS/MS) was performed to identify the absorbed components of toad venom in rat plasma. In vitro experiments were determined to evaluate the therapeutic effects of bufalin and cinobufagin and screen the optimal ratio between them. An in vivo HepG2 tumor-bearing nude mice model was established, and a series of pharmacodynamic indicators were determined, including the body weight of mice, tumor volume, tumor weight, and histopathological examination of tumor. Further, the entire metabolic alterations in tumor after treating with bufalin and cinobufagin were also profiled by UHPLC-MS/MS. Twenty-seven active components from toad venom were absorbed in rat plasma. We found that the cotreatment of bufalin and cinobufagin exerted significant antitumor effects both in vitro and in vivo, which were reflected in inhibiting proliferation and inducing apoptosis of HepG2 cells and thereby causing cell necrosis. After cotherapy of bufalin and cinobufagin for twenty days, compared with the normal group, fifty-six endogenous metabolites were obviously changed on HepG2 tumor-bearing nude mice. Meanwhile, the abundance of α-linolenic acid and phenethylamine after the bufalin and cinobufagin intervention was significantly upregulated, which involved phenylalanine metabolism and α-linolenic acid metabolism. Furthermore, we noticed that amino acid metabolites were also altered in HepG2 tumor after drug intervention, such as norvaline and Leu-Ala. Taken together, the cotreatment of bufalin and cinobufagin has significant antitumor effects on HepG2 tumor-bearing nude mice. Our work demonstrated that the in-depth mechanism of antitumor activity was mainly through the regulation of phenylalanine metabolism and α-Linolenic acid metabolism.

## 1. Introduction

Hepatocellular carcinoma (HCC), the sixth most common cancer and the third leading cause of cancer death in the world in 2020, remains a major challenge to global public health [1]. Surgical resection is still the primary treatment for HCC patients diagnosed at the early stage, but it is not ideal for HCC patients diagnosed at the advanced stage [2]. Currently, despite the use of sorafenib, which can prolong the survival rate, the concomitant problems also greatly affect the prognosis of patients, such as severe side effects, short-term effects, and drug resistance [3]. Traditional Chinese medicine (TCM) has attracted increasing interest as a potential antitumor drug repository due to its beneficial effects and relatively low toxicity [4].

Toad venom, also named as *Chansu*, is a dried product of secretion from *Bufo bufo gargarizans* Cantor or *Bufo melanostictus* Schneider, which was firstly recorded in *Yao Xing Lun,* dated back to the Tang Dynasty, for the treatment of brain dystrophy [5]. According to the Chinese Pharmacopoeia (2020 edition), toad venom possesses a series of pharmacological activities on detoxification and analgesia and has a cardiotonic effect, which is mainly applied for the treatment of ulcer, sore throat, sunstroke, fever, abdominal pain, vomiting, and diarrhea. With the in-depth study of toad venom, it could be considered to be a promising anticancer candidate due to its powerful activity against various cancer cells [2]. Previous phytochemical research revealed that bufadienolides are the main active component of toad venom, such as bufalin and cinobufagin [6]. Bufadienolides, acting as Na^+^/K^+^-ATPase inhibitors, can trigger intracellular Na^+^/Ca^2+^ exchange, increase intracellular calcium levels, and subsequently lead to an increase in cardiac contractility. However, in recent years, bufadienolides have received increasing attention for their antitumor activity. It has been demonstrated that bufalin (Buf) can inhibit HepG2 cell proliferation, invasion, migration, and adhesion, which have a definite effect on HCC [7]. Moreover, the HepG2 cell apoptosis induced by Buf was associated with an increase in Fas, Bax, and Bid expression and a decrease in Bcl-2 expression [8]. Cinobufagin (Cino), another major component of toad venom, has been used in clinical treatment of heart failure, sores, and pain [9]. Previously, we have demonstrated that Cino could effectively inhibit the growth of breast cancer in vitro and in vivo [10]. In addition, Cino exerted effective therapeutic activity against various cancers, such as liver cancer, lung cancer, melanoma, and colon cancer [9,11,12]. Notably, our research has also found that both Buf and Cino can effectively interact with HepG2 cells and be detected intracellularly, providing the basis of its antitumor activity [13]. Jinghui Zhang et al. demonstrated that the combined action of Buf and Cino acted synergistically in inducing apoptosis and inhibiting growth on HepG2 tumor-bearing nude mice [14]. Therefore, it is of great value to study these two active components of toad venom and their pharmacological activities. So far, the underlying mechanisms of toad venom for treating HCC have not been fully implemented, especially the changes in endogenous small molecules after treatment.

Metabolomics provides an effective approach to identify endogenous metabolite alterations and elucidate potential therapeutic mechanisms so as to reflect the overall functional state of the body. This is consistent with the integrity of TCM theory and presents the whole state of human body as under the influence of multiple factors [15]. Currently, metabolomics has a great potential for identifying and evaluating biomarkers and therapeutic targets of various cancers in clinical practice, such as HCC, lung cancer, colorectal cancer, prostate cancer, ovarian cancer, and other diseases [16]. In the present study, we identified the major components of toad venom in rat plasma. Actually, the compounds absorbed into plasma were considered biologically active and deserved further investigation [17]. On the basis of our previous study, we focused specifically on elaborating the antitumor activity and the possible mechanisms of Buf and Cino on HepG2 tumor-bearing nude mice. Concretely, nontargeted tumor metabolomics analysis was performed on ultraperformance liquid chromatography with tandem mass spectrometry (UHPLC-MS/MS) before and after cotreatment of Buf and Cino. Finally, this study will hopefully provide a novel insight to the potential mechanism of toad venom against HCC.

## 2. Results

### 2.1. In Vivo Compound Identification of Toad Venom in Rat Plasma

The major compounds of toad venom in rat plasma were detected using the UHPLC-HR-Q-Exactive Orbitrap MS system. The compound identification was performed based on chromatographic peaks, retention time, potential molecular formula, fragment ion information, and reference data. Previously, a total of 93 compounds in toad venom were characterized in the positive ion mode of the UHPLC-MS/MS system [13]. In this study, 27 compounds of toad venom extract in rat plasma were identified, including 19 prototype compounds and 8 metabolites. The detected prototype compounds included 16 bufogenins, 2 amino acids, and 1 alkaloid. Total ion chromatograms in the positive ion mode of blank plasma, toad venom plasma, and toad venom exact are shown in Figure 1. The details of the 27 identified compounds, including the retention time, chemical formula, measured mass, MS2 fragments, and chemical structure, are presented in Table 1 and Appendix A and Figure 2.

### 2.2. In Vitro Antitumor Effect of Buf and Cino

In this part, Buf and Cino were selected as model drugs to evaluate the anti-HCC activity of toad venom. Firstly, the in vitro antiproliferation activity of Buf and Cino alone or in combination were evaluated via CCK8 assay. As illustrated in Figure 3A,B, the cell viability of HepG2 cells were decreased with the increased drug concentration, showing apparent dose-dependent effects. The half-maximal inhibitory concentration (IC50) of the Buf and Cino were 26.753 μM and 86.025 μM, respectively. Subsequently, the combined therapeutic effect of Buf and Cino was also investigated (Figure 3C). According to quantitative analysis of the Chou–Talalay method, the CI values of Buf/Cino at molar ratios of 2:1, 1:1, and 1:2 were 0.871, 0.760, and 0.845, respectively. These data intuitively indicated that Buf/Cino with a smaller CI value has a stronger antitumor effect in vitro. Compared with the individual drugs, the combined in vitro antitumor efficacy was significantly increased. Therefore, to achieve a maximized effect, the combination of Buf and Cino with a molar ratio of 1:1 was chosen as the optimal regimen for further study. Finally, Hoechst 33342 staining showed a decrease in the number of cells in the treatment group and confirmed the production of apoptotic bodies, which proved that Buf and Cino had proapoptotic activity on HepG2 cells (Figure 3D).

### 2.3. In Vivo Antitumor Effect of Buf and Cino

To verify the antitumor effect of Buf and Cino in vivo, HepG2 cells’ tumor-bearing nude mice model was established. The whole experimental design and protocol are illustrated in Figure 4A. On day 20, the tumor tissues and main organs (heart, liver, spleen, lung, and kidney) of nude mice were isolated and weighed. Tumor volume of the mice treated with NS increased rapidly, showing no inhibitory effects on tumor growth. Compared with the NS group, the tumor growth of nude mice injected with Buf and Cino was statistically inhibited over time (Figure 4B,E). Notably, the tumor inhibition rate of the Mix group was 88.49% (Figure 4D). Meanwhile, there was no obvious decrease in body weight of nude mice, suggesting a suitable dosage during the whole treatment (Figure 4C). The H&E staining of major organs further verified the safety of drug treatment. All the organs had intact cell morphology, without nuclear shrinkage, cytoplasmic lysis, and inflammatory cell infiltration (Figure 5). As shown in Figure 6, the tumor cells in the NS group had the complete morphology, with clearly visible nuclei and no nuclear pyknosis. However, after the administration of Buf and Cino, the tumor cells became scattered and exhibited varying degrees of necrosis, such as nuclear fragmentation, nuclear dissolution, and even the disappearance of the nucleus. The proliferation and apoptosis of tumor cells were also evaluated by Ki-67 staining and TUNEL staining, respectively. Results demonstrated that the decreased positive Ki-67 expression and increased apoptosis rate of tumor cells were obviously observed compared with the NS group (Figure 6).

### 2.4. Nontargeted Metabolomics Analysis of Tumor Tissues

#### 2.4.1. Stability Test of Tumor Metabolomic Method

Firstly, 20 μL of each sample was mixed into QC samples and detected five times. On PCA analysis, the distribution of QC samples with positive and negative modes were closely concentrated, suggesting a good experimental repeatability and stable detecting system (Figure 7E and Appendix A). Further, Pearson correlation analysis was performed on QC samples, as shown in Figure 7D and Appendix A. The experimental results showed that the correlation coefficients between QC samples were above 0.9, indicating that the experimental repeatability is good. Furthermore, Hotelling’s T2 test, Multivariate Control Chart (MCC), and relative standard deviation of ion peak abundance in QC samples were performed, which also reflected the true and reliable data with the detecting system (Figure 7A–C and Appendix A).

#### 2.4.2. Identification of Differential Metabolites

The structures of metabolites in tumor tissue samples were identified by matching the retention time, molecular mass (molecular mass error within <10 ppm), second-order fragmentation profile, collision energy, and other information of the metabolites in the local database. Ultimately, 621 metabolites were identified, including 363 metabolites in the positive ion mode and 258 metabolites in the negative ion mode. Among them, the proportions of lipid and lipid-like molecules, organic acids, and benzenoids were about 30%, 18%, and 13%, respectively. All metabolites detected in positive and negative ion mode were differentially analyzed based on univariate analysis. The metabolites with FC > 1.5 or FC < 0.67 and *p*-value < 0.05 between two groups were visualized in the form of a volcano diagram (Figure 8A). Further, the unsupervised 3D-PCA was applied to assess the overall distribution among all samples and the stability of the entire analysis process. As shown in Figure 8B, the two groups of samples were well-separated, suggesting a large number of different types of metabolites that presented between Mix and NS. Subsequently, the supervised PLS-DA were established to verify the difference between the two groups. As illustrated in Figure 8C, there was the satisfactory separation on the PLS-DA model between the NS and Mix groups. The values of R2Y and Q2 in the PLS-DA model between Mix and NS were 0.994 and 0.57, respectively (Figure 8D). Additionally, the data performed by permutation test revealed that the obtained model exhibited no overfitting behavior. Differential metabolites were screened with the requirement of VIP > 1 and *p* < 0.05 in the test. Totally, 89 potential differential metabolites were obtained, including 20 differential metabolites between Buf and NS, 13 differential metabolites between Cino and NS, and 56 differential metabolites between Mix and NS. The detailed information of these differential metabolites is presented in Appendix A.

#### 2.4.3. Hierarchical Cluster Analysis of Differential Metabolites

The expression of differential metabolites was calculated by distance matrix and analyzed by hierarchical cluster to comprehensively and visually obtain the differences in the expression patterns of metabolites in different samples. The clustering heatmap analysis is shown in Figure 9B. Compared with the NS group, the expression of 50 species of metabolites in the Mix group were upregulated, while 6 species of metabolites were downregulated (see Appendix A). The tumor metabolomics results revealed that the abundance of metabolite species in nude mice tumor changed significantly after the administration of Buf and Cino. When compared with the NS group, the differential metabolites in the Mix group intervened on by Buf and Cino showed an upregulated trend, including phenylethylamine, linolenic acid, norvaline, L-serine methyl ester, L-leucyl-l-leucine methyl ester, and methyltyrosinate; also, differential metabolites exhibited a downregulated trend, such as Carbobenzyloxy-l-norvalyl-l-norleucine, 1,2-dioleoyl-sn-glycero-3-phosphatidylcholine, 1-stearoyl-2-linoleoyl-sn-glycerol, 7-hydroxymitragynine, 3-hydroxybenzaldehyde, and Pg 42:11. Figure 9A displays the correlation analysis of differential metabolites between the Mix and NS groups. Red color indicates a positive correlation, while blue color indicates a negative correlation. The higher value represents the closer correlation between two metabolites.

#### 2.4.4. Pathway Analysis of Differential Metabolites

Differential metabolites pathway enrichment analysis is of great significance for revealing the pathogenesis of diseases. In this study, to explore the metabolic pathways associated with the efficacy of Buf and Cino in the treatment of HCC, the differential metabolites were imported into a MetaboAnalyst 5.0 database, and pathways with impact value > 0.1 were marked as potential target pathways. As a result, compared with the NS group, differential metabolite pathways of Buf and Cino used individually were both involved in nicotinate and nicotinamide metabolism (Figure 10A). Meanwhile, phenylalanine metabolism and α-linolenic acid metabolism were closely related to the therapeutic effects of Buf and Cino against HCC. As illustrated in Figure 10B,C, the differential metabolic pathways were colored and displayed by using the KEGG pathway mapper function according to the upregulated and downregulated information. On the one hand, phenylalanine could regulate substance metabolism as well as be a mediating factor of apoptosis. It is essential for the production of the nonessential tyrosine, the process of which was firstly discovered by Womack and Ross in 1934 [18]. This conversion is catalyzed by phenylalanine hydrolase, which is primarily active in the liver. Tyrosine and phenylalanine levels were decreased in the serum of HCC patients [19]. Moustafa R.K. et al. found that the pathway that allows the conversion of phenylalanine to tyrosine may lead to the accumulation of phenylalanine, which further leads to mitochondria-mediated apoptosis and thereby causes cancer cell death [20]. In this paper, both in vitro and in vivo assays have demonstrated the HepG2 cell apoptosis induced by the cotreatment of Buf and Cino. Therefore, we may speculate that the increased level of phenylalanine in HCC tumor after Buf and Cino intervention may be closely related to the decreased activity of phenylalanine hydrolase. On the other hand, in the fatty acid metabolic pathway, fatty acid synthesis is an important process that converts nutrients into metabolic intermediates for cell biosynthesis, energy storage, and signal transduction [21]. Cell proliferation is a general feature of all cancerous cells and requires fatty acids to synthesize signal transduction molecules and tumor cell membranes [22]. During the fatty acid synthesis process, fatty acid synthase plays a key role. The expression level of fatty acid synthase in cancer cells was significantly higher than that in normal cells [23]. A study had shown that α-linolenic acid, a polyunsaturated fatty acid, can inhibit the proliferation and metastasis of osteosarcoma cells by inhibiting the expression of fatty acid synthase [24]. Our results showed that α-linolenic acid was lowly expressed in HCC tumors, while Buf and Cino intervention increased its levels. Results of in vitro and in vivo experiments indicated that Buf and Cino have a notable effect on inhibiting HepG2 cell proliferation and inducing HepG2 cell apoptosis. Taken together, the anti-HCC effect of Buf and Cino may be attributed to the downregulation of fatty acid synthase activity and disorder of fatty acid synthesis.

## 3. Discussion

HCC is one of the severe malignancies that causes death in humans [25]. Screening high-efficiency and low-toxicity anti-HCC drugs from TCM has always been a research hotspot. TCM with multiple components, multiple efficacies, and multiple targets regulates the state of the body as a whole so as to play a therapeutic role. Since most oral TCMs have the first-pass effect, only components that are absorbed into the blood can function effectively [26]. Toad venom has a long history in the treatment of HCC, the main antitumor components of which are bufadienolides [5]. According to previous studies, Buf and Cino were considered as the main antitumor components in toad venom, the respective anticancer effects of which were closely linked with the downregulation of prosurvival proteins (Bcl-2) and upregulation of the proapoptotic proteins (Fas and Bax) [14,27]. Up to now, the anticancer mechanisms of Buf and Cino were evaluated individually, which were insufficient to guide their combined clinical use. Therefore, the mechanism of the combined effect of Buf and Cino should be further emphasized.

During this study, the absorbed components of toad venom in rat plasma were identified by using the UHPLC-HR-Q-Exactive Orbitrap MS system to better screen bioactive components. As mentioned previously, these two components played an important role in the treatment of HCC by toad venom. In our current study, the optimal ratio of Buf and Cino was selected as 1:1 via CCK-8 assay. Results of in vitro studies found that the combination therapy of Buf and Cino showed effective action on the inhibition of HepG2 cell proliferation and increase in HepG2 cell apoptosis, exhibiting an enhanced in vitro antitumor activity compared with individual treatment. Therefore, we explored the perturbation of endogenous metabolites in tumor tissue after the cotreatment of Buf and Cino, hoping to explain the in-depth mechanism of toad venom against HCC. After the intervention of Buf and Cino, the tumor growth of the Mix group was significantly inhibited. No obvious toxicity was observed in H&E staining results of major organs. An in vivo antitumor mechanism of Buf and Cino showed that Buf and Cino could inhibit cell proliferation and induce cell apoptosis on HepG2 tumor-bearing nude mice. Further, the tumor metabolomics results found that the abundance of many metabolite species in nude mice tumor changed significantly after the administration of Buf and Cino. The pathway enrichment of differential metabolites between the Mix group and the NS group were mainly involved in phenylalanine metabolism and α-linolenic acid metabolism.

It has been shown that the phenylalanine metabolism pathway was altered in gastric cancer and prostate cancer [28,29]. According to our studies, the biomarker that participated in phenylalanine metabolism in HCC was phenethylamine, which is an aromatic amino acid affecting the pathology of hepatitis B [14]. For example, phenylalanine metabolism efficiency decreased in the gut of patients with early chronic hepatitis B [30]. Moreover, phenylalanine could regulate substance metabolism as well as be a mediating factor of apoptosis. The induction of apoptosis by phenethylamine may be involved in three mechanisms: the FasR-mediated cell death receptor pathway, the Rho/ROCK pathway to activate mitochondria mediated apoptosis, and LAMP2 in graninase B signaling-mediated apoptosis [31]. Lamin B1 and PAK1 were found in the Fas/Fas ligand death receptor pathway, which have previously been shown to be involved in apoptosis [20]. Lamin B1, as the main component of the subnuclear layer, plays an important role in maintaining the integrity of the nuclear membrane. Destruction of nuclear membrane integrity is a sign of apoptosis. During the apoptosis process, Lamin B1 mRNA levels were decreased, which may be due to the induction of p53 or pRB tumor suppressor pathways [32]. References also indicated that downstream effector PAK1 of Fas/Fas ligand complex prevents apoptosis by restricting the expression of proapoptotic proteins or regulating post-translational modification of effectors [20]. Moustafa R. K et al. further confirmed that the altered phenylalanine metabolism activated several apoptosis pathways associated with key proteins such as HADHA and ACAT1 [20]. In view of the upregulation of phenethylamine in HCC with Buf and Cino cotreatment, an induction on cell apoptosis capacity can be speculated, which is consistent with the apoptotic phenotype observed in Buf and Cino-treated cells and tumor tissues.

α-linolenic acid metabolism is an extremely vital metabolic pathway that interferes with tumor proliferation and metabolism [33,34]. α-linolenic acid, a polyunsaturated fatty acid, has a variety of benefits, including improved immunity and anticancer and anti-inflammatory effects [35]. Due to its anti-inflammatory and antitumor properties, α-linolenic acid has been reported to be a potential adjuvant therapy candidate for breast, pancreatic, and colon cancer [36]. Currently, the main attention on the possible mechanism of α-linolenic acid inhibiting the growth of different breast cancer cell lines is increasing lipid peroxidation, altering the expression and activation of cell membrane receptors, changing transcription factors, and regulating tumor suppressor genes such as phosphatase and tensin homologous genes [37]. A study has shown that α-linolenic acid regulates the growth of cancer cells in breast cancer (MCF-7 and MDA-MB-231) and cervical cancer (SiHa and HeLa) by reducing NO release and inducing lipid peroxidation [38]. Julie K. et al. found that α-linolenic acid can reduce HER2-overexpressing breast cancer growth and demonstrated for the first time that docosahexaenoic acid (DHA) is responsible for the effects of α-linolenic acid-rich diets on HER2 signaling pathways [39]. Wang et al. found that α-linolenic acid could inhibit the migration of human triple-negative breast cancer cells by reducing the expression of Twist1 and inhibiting Twist1-mediated epithelial mesenchymal transformation [35]. Li et al. reported that dietary supplementation with α-linolenic acid induced n-3 LCPUFAs conversion and reduced prostate cancer growth in a mouse model [33]. Interestingly, the content of α-linolenic acid in HCC was lower than that of normal tissues [40]. In HepG2 cells, α-linolenic acid suppressed the proliferation, migration, and invasion. An evaluation of the underlying mechanisms suggested that α-linolenic acid enhances FXR expression and thus inhibits the Wnt/β-catenin signaling pathway, thereby hinting HCC progression [40]. Furthermore, Cis9 and trans11 conjugated linoleic acid induce HepG2 cell apoptosis by activating PPAR-γ signaling pathway [41]. Vecchini et al. reported that dietary intake of α-linolenic acid could decrease the expression of COX-2 and increase the apoptosis of liver cancer cells [42]. In this study, the abundance of α-linolenic acid detected in HepG2 tumor treated with Buf and Cino was significantly increased compared with the NS group, indicating their potential contribution to the progression of HCC.

Meanwhile, we noticed that amino acid metabolites were also altered in HepG2 tumor-bearing nude mice after the cotreatment of Buf and Cino, such as norvaline and Leu-Ala. Among them, norvaline, an isomer of valine, was involved in the cytotoxic activity of macrophages against breast tumor cells [43]. Furthermore, previous reports showed that norvaline could interact with gut microbiota to affect the occurrence of colorectal cancer [44]. A clinical study has confirmed that the levels of norvaline and Leu-Ala in plasma were significantly lower in patients with liver cancer than in healthy people. Compared with the NS group, the levels of these two amino acids were recovered after cotreatment of Buf and Cino, which is consistent with previous studies. Additionally, it is worth considering that some metabolites without ionization cannot be detected by UHPLC-MS/MS. So, in the future, nuclear magnetic resonance technology will be applied to make up for the deficiency of mass spectrometer detection [3]. Due to the limitation of the KEGG database and the MetaboAnalyst 5.0 database, several identified differential metabolites could not be matched with the corresponding KEGG ID, so they failed to participate in pathway enrichment, resulting in a small number of enriched pathways. Summarily, using Buf and Cino as model drugs, this study confirmed the antitumor activity of toad venom in HepG2 tumor-bearing nude mice and revealed its antitumor mechanism at a metabolic level. In our future work, transcriptomics and proteomics will be involved in deeper mechanism research of toad venom against HCC.

## 4. Materials and Methods

### 4.1. Materials and Reagents

Dried toad venom secretion originated from *Bufo melanostictus* Schneider was provided by Beijing newborn toad breeding center (Beijing, China) and authenticated as toad venom by Professor Qingrong Pu from the affiliated traditional Chinese medicine hospital of Southwest Medical University (Luzhou, China). Bufalin and cinobufagin were provided by Chengdu Ruifensi Biotechnology Co., LTD. Methanol, acetonitrile, and formic acid were purchased from Thermo Fisher Scientific (Shanghai, China) Co., Ltd. Cell-Counting-Kit-8 tests were obtained from Beyotime Biotechnology Co., Ltd. (Shanghai, China). Other reagents used in this study were of analytical reagent grade.

### 4.2. Cells and Animals

HepG2 cells were provided by Professor Jie Qing from the affiliated traditional Chinese medicine hospital of Southwest Medical University (Luzhou, China) and cultured in a DMEM high-glucose medium at 37 °C in an atmosphere containing 5% CO_2_. The female BALB/c nude mice (five weeks old) were purchased from Chongqing Tengxin Biotechnology Co., LTD (Chongqing, China). The nude mice were housed in a specific-pathogen-free (SPF) cleanliness level barrier system with a room temperature of 25 ± 2 °C and a relative humidity of 50 ± 10%. Animal welfare and experimental procedures were strictly performed in accordance with the Guide for the Care and Use of Laboratory Animals and the related ethics regulations of Southwest Medical University. Animal protocols were reviewed and approved by the Internal Animal Care and Use Committee of Southwest Medical University (Approval Number: 20211115-010).

### 4.3. In Vivo Compound Identification of Toad Venom in Rat Plasma

The sample preparation and UHPLC-MS/MS detection conditions are presented in the Appendix A. Based on the standard substances database established in our previous research, the prototypical compounds of toad venom in rat plasma were identified. Combined with reported references, metabolites of toad venom were further identified according to the first and second mass spectra of prototypes. The compounds absorbed in plasma were considered biologically active and deserved further study [45].

### 4.4. Cell Counting Kit-8 (CCK-8) Assay

The CCK-8 assay was used to study the antiproliferation of Buf and Cino. Briefly, HepG2 cells were grown in 96-well plates at a density of 1 × 10^5^ per well prior to the treatment with various concentrations of Buf (1.563, 3.125, 6.25, 12.5, 25, 50, 100, 200, 400 nM), Cino (12.5, 25, 50, 100, 200, 400, 800, 1600 nM), and Buf/Cino (2:1, 1:1, 1:2) for 24 h. Subsequently, the culture medium was removed and replaced by 10 μL CCK-8 reagent for an additional 2 h incubation. Finally, the absorbance (Abs) of each well was measured at 450 nm using a microplate reader (Synergy2, Biotek, Highland Park, IL, USA). The cell viability was calculated. The combination index (CI) was calculated according to the Chou–Talalay method, used for quantitative analysis of combination therapy in vitro [46].
Cell viability = [(A_M_ − A_B_)/(A_C_ − A_B_)]

A_M_, A_C_, and A_B_ are defined as the absorbances of the drug-treated group, control group, and blank group, respectively.


$$\mathrm{CI}=\frac{(D_{50})A}{\left(D\right)A}+\frac{(D_{50})B}{\left(D\right)B}$$


where (D) A and (D) B are the concentration of A and B used in combined therapy to achieve 50% inhibitory effects; (D_50_) A and (D_50_) B are the concentration of A and B to achieve the same effects individually.

### 4.5. Hoechst 33342 Staining Assay

HepG2 cells were grown in 6-well plates at a density of 2.5 × 10^5^ per well for 24 h. The culture medium was removed and replaced by Buf (12.5 nM), Cino (12.5 nM), and Buf/Cino (12.5/12.5 nM), respectively. After 24 h incubation, each well was added with 4% formaldehyde solution for the 30 min of fixation at −20 °C. After being washed twice with phosphate buffer, HepG2 cells were stained with Hoechst33342 solution for 10 min, then discarded and washed three times. Finally, a fluorescence microscope (Evos, Life Technologies, Carlsbad, CA, USA) was used to observe cell apoptosis.

### 4.6. In Vivo Antitumor Efficacy

To obtain tumor-bearing nude mice, the female BALB/c nude mice were injected with HepG2 cells (about 1 × 10^7^ cells) into the right armpit. When the volume of tumors reached about 100 mm^3^, nude mice were divided randomly into four groups (seven mice per group) as follows, (1) Normal Saline (NS, ip every two days, same dosing volume as Buf group), (2) Buf (ip every two days, 1 mg/kg), (3) Cino (ip every two days, 1.14 mg/kg), and (4) Mix (Buf (ip, 1 mg/kg) plus Cino (ip, 1.14 mg/kg)). Tumor volume and body weight of nude mice were monitored every two days during 20-day treatment. At day 20, the nude mice were sacrificed by cervical dislocation. The tumor volume (V) and the tumor inhibition rate (TIR) were calculated, respectively. The excised organs were used to evaluate the safety of the therapy. Pathological examinations, including H&E staining, Ki-67 staining, and TUNEL staining, were performed on the tumor tissues. The detailed procedures are shown in the Appendix A.
V = 0.5 × length × wide^2^
TIR = [(W_C_ − W_E_) / W_C_] × 100%
where W_E_ and W_C_ are defined as the tumor weight of the experiment group and control group.

### 4.7. Nontargeted Metabolomics Analysis of Tumor Tissues

#### 4.7.1. Metabolites Extraction

Firstly, 80 mg tumor tissue was weighed and slowly thawed at 4 °C. After that, the sample was immersed in 200 μL water to homogenize and treated with 800 μL precooled methanol/acetonitrile solution (1:1, *v*/*v*) to sonicate for 30 min at 4 °C. Then, it was placed at −20 °C for 2 h to precipitate protein. Subsequently, after centrifuging for 20 min at 4 °C at the rate of 14,000 rpm, the supernatant was taken and freeze-dried. Finally, 100 μL acetonitrile/water solution (1:1, *v*/*v*) was added to redissolve and vortex for 1 min. After centrifugation again at 14,000 gpm for 15 min at 4 °C, the supernatant was taken for UHPLC-Q-Exactive Orbitrap MS analysis.

#### 4.7.2. UHPLC-Q-Exactive Orbitrap MS Analysis

In this study, UHPLC-Q-Exactive Orbitrap MS analysis was conducted with the application of the ThermoFisher Scientific Vanquish Ultra-High Performance Liquid Chromatography system associated with the ThermoFisher Scientific Q-Exactive Orbitrap mass spectrometer. The chromatographic separation was performed on the Xbridge BEH HILIC column (2.1 × 100 mm, 2.5 μm). The temperatures of the sample disk and column oven were set at 4 °C and 25 °C, respectively. The mobile phase consisted of solvent A (water containing 25 mM ammonium acetate and 25 mM ammonia solution) and B (acetonitrile). At a 0.3 mL/min flow rate, the injection volume was kept at 2 μL. The gradient elution procedure was as follows: 0–1.5 min, 98% B, 1.5–12 min, 98–2% B, 12–14 min, 2% B, 14–14.1 min, 2–98% B, and 14.1–17 min, 98% B. In order to avoid the influence of instrument detection signal fluctuation, random sequence was used for continuous analysis of samples. Quality control (QC) samples were inserted into the sample queue to monitor and evaluate the stability of the system and the reliability of the experimental data.

Further, the samples were analyzed by Q-Exactive Orbitrap mass spectrometer. Both the primary and secondary spectrograms of the samples were collected by electrospray ionization (ESI) in positive and negative ion modes. ESI source and mass spectrum setting parameters were as follows: atomizing gas auxiliary heating 1 (Gas1); 60, auxiliary heating 2 (Gas2); 60, curtain gas (CUR); 30 psi, ion source temperature; 600 °C, spray voltage (ISVF); ±5500 V (positive and negative two modes). Primary mass charge ratio detection range: 80–1200 Da; resolution: 60,000; scanning accumulation time: 100 ms. The secondary stage adopted the segmental acquisition method, with a: scanning range: 70–1200 Da; secondary resolution, 30,000; scanning accumulation time: 50 ms; dynamic exclusion time: 4 s.

#### 4.7.3. MS Data Processing and Analysis

The original data were converted into MzXML format by ProteoWizard software. XCMS software was used for peak alignment, retention time correction, and peak area extraction. PCA and PLS-DA were performed via SIMCA-14.1 software. The screening condition of differential metabolites were VIP > 1 and *p* < 0.05 in the *t*-test, which was chosen as the in-depth metabolic pathway analysis. Eventually, pathway enrichment was obtained from the MetaboAnalyst 5.0 (https://www.metaboanalyst.ca, 10 September 2022) platform. All the data were expressed as mean ± standard deviation (SD). The independent sample’s *t*-test of two groups was performed by SPSS 26.0 (SPSS Inc., Chicago, IL, USA).

## 5. Conclusions

The current work identified the active components of toad venom in rat plasma. Next, we conducted in vitro and in vivo experiments to explore the underlying mechanism of Buf and Cino against HCC. The antitumor effect after the cotreatment of Buf and Cino was mainly through regulating the phenylalanine metabolism and α-Linolenic acid metabolism. It is suggested that amino acid metabolites such as norvaline and Leu-Ala may play the important role regarding toad venom in treating HCC. Taken together, these results suggest the use of toad venom as a promising candidate for further clinical studies against HCC.

## Figures and Tables

**Figure 1 molecules-27-07758-f001:**
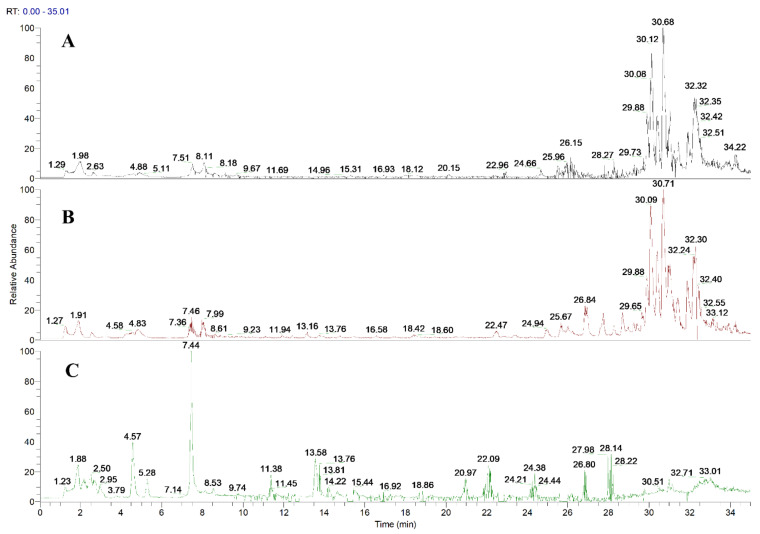
Identification of active components of toad venom in rat blood. (**A**) Blank blood. (**B**) Toad venom in rat blood. (**C**) Toad venom extract.

**Figure 2 molecules-27-07758-f002:**
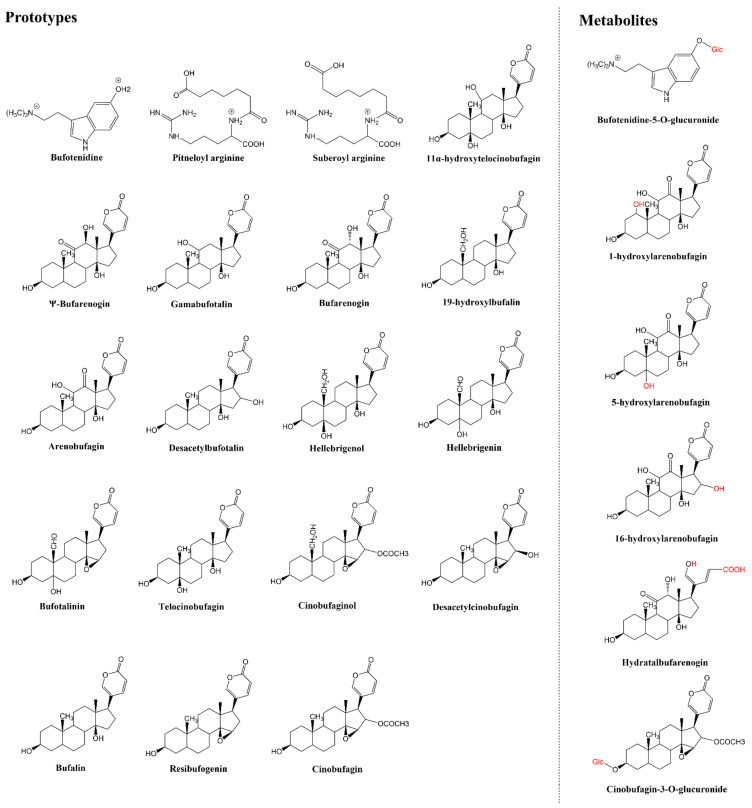
The detailed chemical structure of identified components of blood in rat plasma.

**Figure 3 molecules-27-07758-f003:**
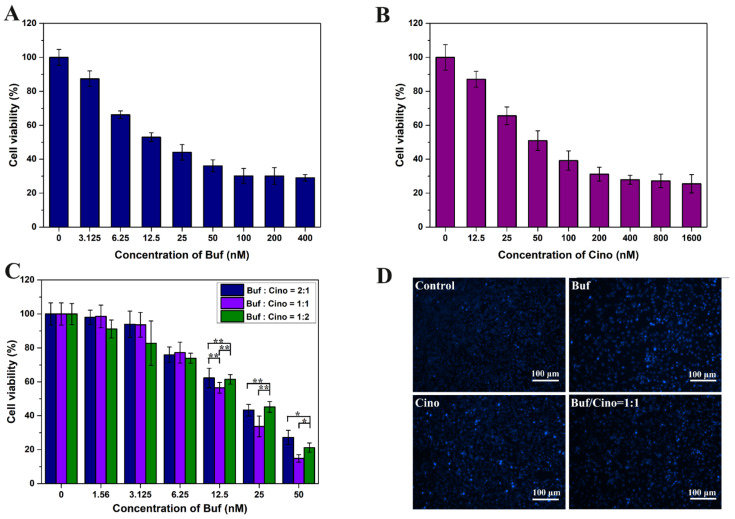
The HepG2 cells viability were cultured in different concentrations of drugs for 24 h. (**A**) Buf, (**B**) Cino, (**C**) used in combination. (**D**) Hoechst 33342 staining was performed by the treatment of Buf, Cino, and the combination, respectively. “*” means *p* < 0.05, and “**” means *p* < 0.01 between two groups.

**Figure 4 molecules-27-07758-f004:**
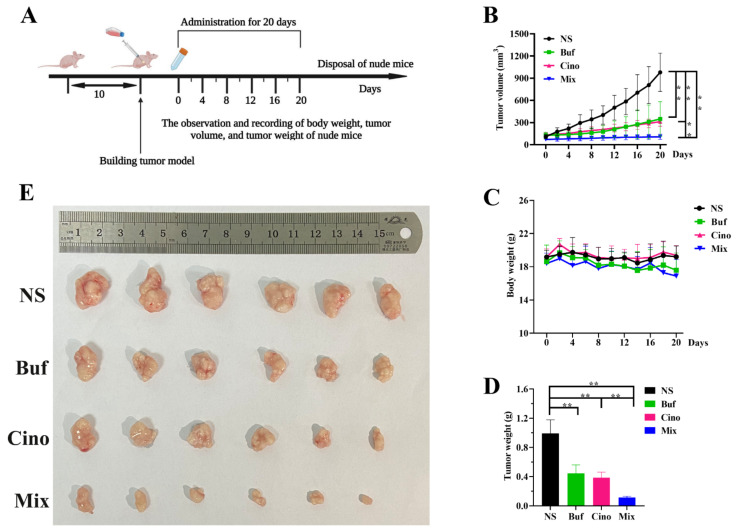
The cotreatment of bufalin and cinobufagin inhibited the HepG2 tumor growth. (**A**) The designed experimental scheme on in vivo nude mice model. (**B**–**D**) Changes in tumor volume, body weight, and tumor weight during 20 days’ drug intervention. (**E**) The excised tumor images with the treatment of NS, Buf, Cino, and in combination for 20 days. Values were presented as the mean ± S.D., *n* = 6. “**” means *p* < 0.01 between two groups.

**Figure 5 molecules-27-07758-f005:**
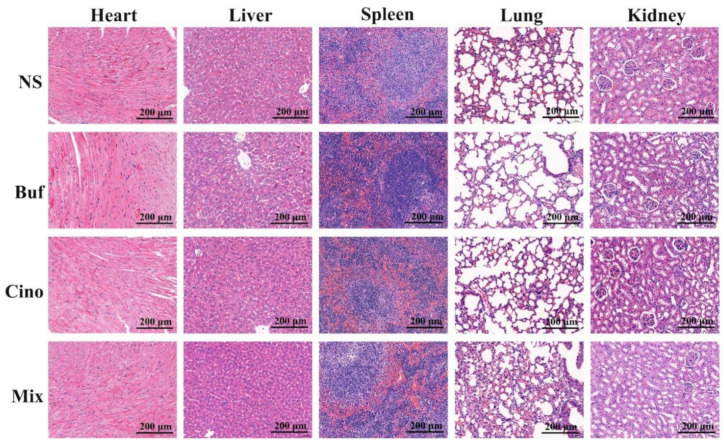
H&E staining analysis of major organs (heart, liver, spleen, lung, kidney) of HepG2 tumor-bearing nude mice model after 20 days’ drug intervention (original magnification: 200×).

**Figure 6 molecules-27-07758-f006:**
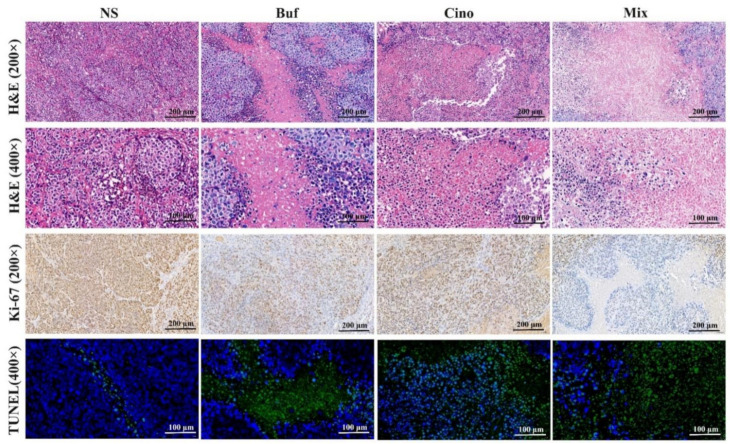
Pathological analysis of tumor with different treatment groups. H&E staining, pale pink areas suggest necrosis area. Ki-67 staining, Brown color represents the nucleus of vigorously proliferating cells. TUNEL staining, green fluorescence represents the nucleus of apoptotic cells. (Original magnification: 200× and 400×).

**Figure 7 molecules-27-07758-f007:**
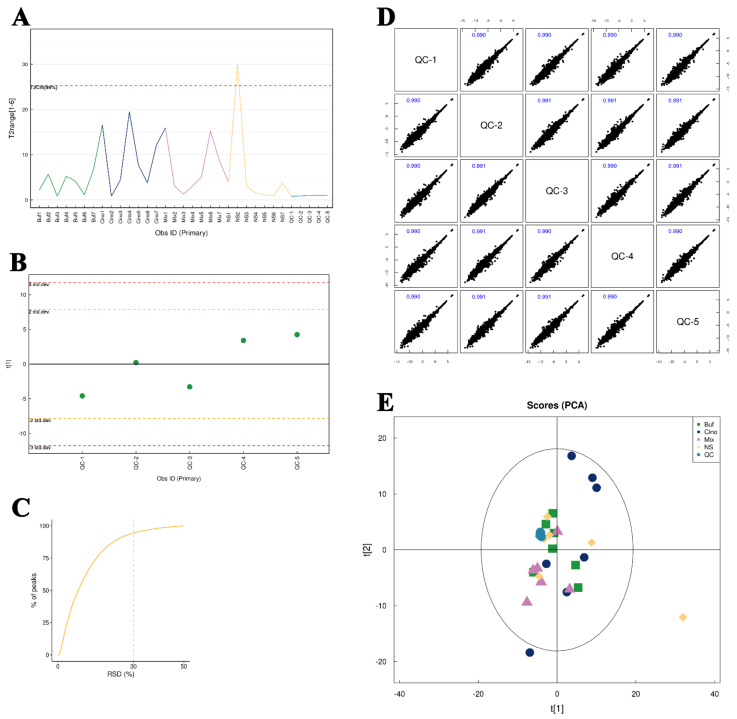
Results of quality control. (**A**) Hotelling’s T2 test for the total sample in positive mode. (**B**) Multivariate control chart of QC samples in positive mode. (**C**) Relative standard deviation of QC samples in positive mode. (**D**) Pearson correlation analysis of QC samples in positive mode. (**E**) Principal component analysis of QC samples in positive mode.

**Figure 8 molecules-27-07758-f008:**
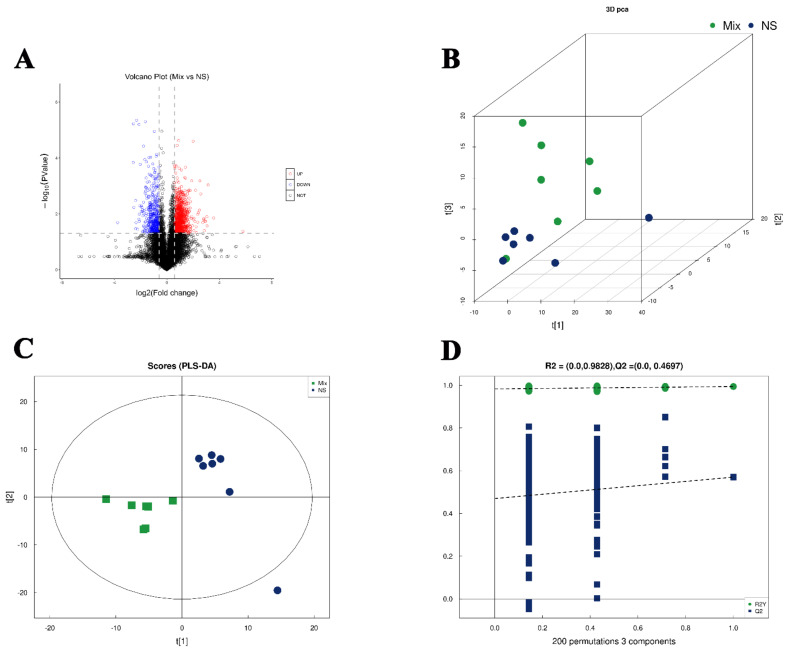
(**A**) Volcanograms showing up- and downregulated metabolites between NS and Mix group. Blue color indicates downregulated metabolites, while red color represents upregulated metabolites. (**B**) Score plot of 3D principal component analysis between NS and Mix group. (**C**) Score plot of partial least-squares discriminant analysis between NS and Mix group. (**D**) Corresponding validation plot from NS and Mix group.

**Figure 9 molecules-27-07758-f009:**
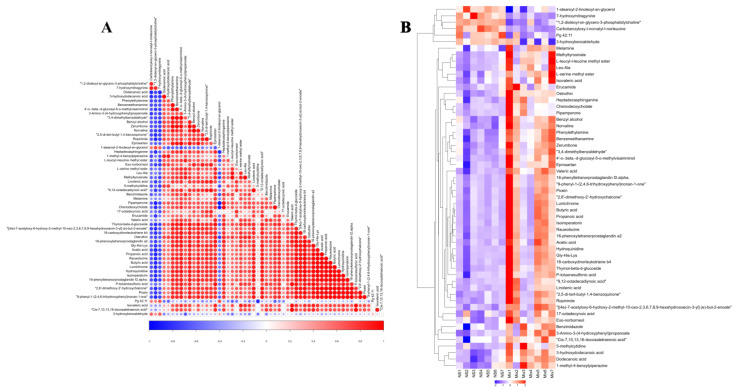
(**A**) Correlation analysis of differential metabolites between NS and Mix group. Red color indicates positive correlation and blue color indicates negative correlation. The higher the value, the stronger the correlation between the two metabolites. (**B**) Heatmap analysis of differential metabolites between NS and Mix group. Red color indicates higher concentration of differential metabolites, blue color indicates concentration of differential metabolites.

**Figure 10 molecules-27-07758-f010:**
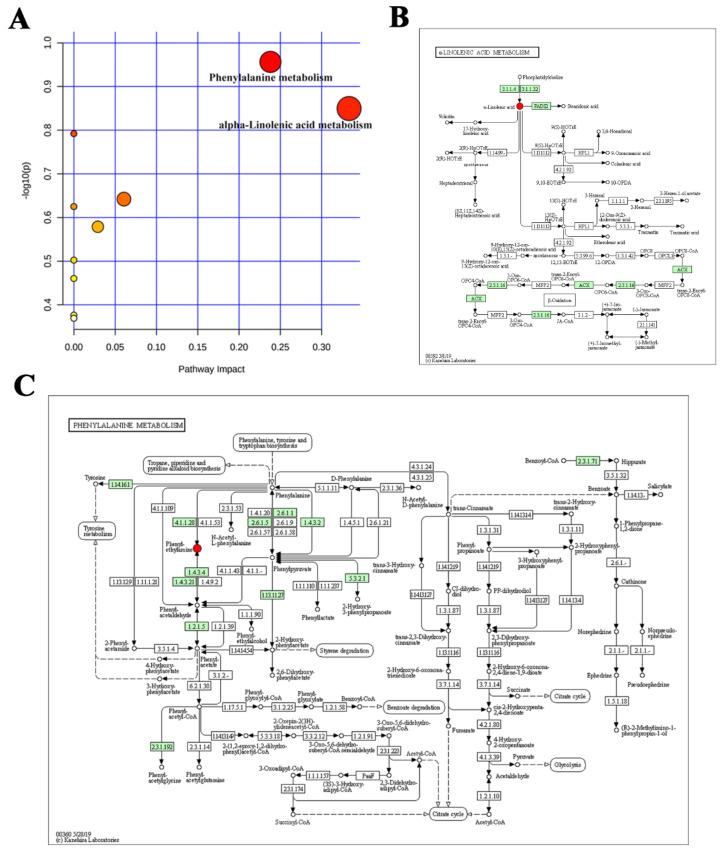
(**A**) Metabolic pathway of differential metabolites between NS and Mix group enriched by MetaboAnalyst 5.0. (**B**,**C**) KEGG enrichment pathway of α-linolenic acid metabolism and phenylalanine metabolism, respectively.

**Table 1 molecules-27-07758-t001:** UHPLC-HR-Q-Exactive Orbitrap MS data of 27 compounds from toad venom extract detected in rat plasma.

No	Compound	Type	RT (Min)	Chemical Formula	Error (Ppm)	[M+H]^+^ Measured Mass	[M+H]^+^ Calculated Mass
1	Bufotenidine-5-O- glucuronide	M	2.54	C_19_H_27_O_7_N_2_	−0.399	395.18112	395.1812
2	Bufotenidine	P	2.62	C_13_H_19_ON_2_	−0.020	219.14917	219.1492
3	Pitneloyl arginine	P	5.28	C_13_H_25_O_5_N_4_	−1.060	317.18185	317.1820
4	Suberoyl arginine	P	7.43	C_14_H_27_O_5_N_4_	−0.533	331.19702	331.1976
5	1-hydroxylarenobufagin	M	9.15	C_24_H_33_O_7_	−1.546	433.22128	433.2148
6	5-hydroxylarenobufagin	M	9.65	C_24_H_33_O_7_	−1.477	433.22131	433.2148
7	16-hydroxylarenobufagin	M	10.01	C_24_H_33_O_7_	−1.200	433.22153	433.2148
8	Hydratalbufarenogin	M	10.06	C_24_H_35_O_7_	−0.490	435.23709	435.2377
9	11α-hydroxytelocinobufagin	P	10.52	C_24_H_35_O_6_	−1.491	419.24234	419.2428
10	Ψ-Bufarenogin	P	10.75	C_24_H_33_O_6_	−1.522	417.22638	417.2272
11	Gamabufotalin	P	11.34	C_24_H_35_O_5_	−1.837	403.24756	403.2479
12	Bufarenogin	P	11.94	C_24_H_33_O_6_	0.539	417.22726	417.2272
13	19-hydroxylbufalin	P	12.43	C_24_H_35_O_5_	−0.101	403.24799	403.2479
14	Arenobufagin	P	12.88	C_24_H_33_O_6_	−1.522	417.22733	417.2272
15	Desacetylbufotalin	P	13.16	C_24_H_35_O_5_	−0.324	403.24774	403.2479
16	Hellebrigenol	P	13.38	C_24_H_35_O_6_	0.035	419.24289	419.2428
17	Hellebrigenin	P	13.76	C_24_H_33_O_6_	−0.132	417.22711	417.2272
18	Bufotalinin	P	15.97	C_24_H_31_O_6_	−1.241	415.21106	415.2115
19	Telocinobufagin	P	22.47	C_24_H_35_O_5_	−0.101	403.24774	403.2479
20	Cinobufaginol	P	24.43	C_26_H_35_O_7_	−0.784	459.23801	459.2377
21	Desacetylcinobufagin	P	26.84	C_24_H_33_O_5_	−0.901	401.23135	401.2323
22	Bufalin	P	26.95	C_24_H_35_O_4_	−1.178	387.25244	387.2530
23	Resibufogenin	P	28.69	C_24_H_33_O_4_	−0.950	385.23697	385.2373
24	Cinobufagin	P	28.89	C_26_H_35_O_6_	−0.265	443.24225	443.2428
25	Unknown	M	29.76	C_17_H_27_O_2_	−1.013	263.20032	263.2084
26	Cinobufagin-3-O- glucuronide	M	31.19	C_31_H_41_O_12_	−0.076	605.45453	605.4545
27	Unknown	M	31.57	C_20_H_33_O_2_	−1.824	305.24695	305.2553

## Data Availability

The Appendix A shows all data produced in the course of this study.

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
