# Peer review of "Uncovering the Mechanisms of Active Components from Toad Venom against Hepatocellular Carcinoma Using Untargeted Metabolomics"

_molecules, 2022, doi:10.3390/molecules27227758_

Round 1

Reviewer 1 Report

In this paper, the metabolism of chemical components in toad skin secretion in mice was identified by LC-MS. The therapeutic effects of bufalin and Cinobufagin on HepG2 tumor were studied. The changes of tumor metabolites treated by bufalin and Cinobufagin were further analyzed. The amount of data in this paper is sufficient, which has certain reference value for the development of anti-tumor drugs using toad active molecules. However, there are still some problems in the paper:

1. At present, there have been many studies on the anti-tumor mechanism of bufalin and Cinobufagin. Although this study failed to study the action targets and signal pathways of these compounds in detail, at least some relevant studies should be cited to explore the correlation between signal pathways and metabonomics in combination with tumor metabonomic data. It is precisely because of the lack of necessary discussion that the whole paper looks like a pile of data and lacks a reasonable explanation of scientific questions. We suggest the authors rewrite some of the Results section and the whole Discussion section.

2. The illustrations in the paper are too simple to allow readers to quickly and comprehensively understand the research results, so it is recommended to supplement.

3. Most of the charts lack statistical analysis.

4. Bars are missing in most of the figures.

5. There is no important control in pharmacodynamics research. For example, positive drug control, Sham group control, etc.

6. As the most important indicator of tumor efficacy, we hope the author can add a mouse survival curve.

7. The sources of some important reagents, such as bufalin and Cinobufagin, need to be supplemented.

8. The author should supplement the ethical approval documents for relevant animal experiments.

Author Response

Thank you for your efficient work and constructive comments. 

Reviewer 2 Report

The heading numbers need to be fixed. Numbers are not sequential.

I would suggest moving some of the ion fragment values to another table and put in the supplemental data. The table is too busy. Needs to be more concise.

Figure 4 graph B since the lines are color coded it does not appear that the lines need to be lettered and the letters are not defined in the figure caption or in the text from what I could tell.

Interesting paper.

Author Response

Thank you for your efficient work and kind consideration. 

Round 2

Reviewer 1 Report

The author responded to the comments of the reviewers. Although the experiments were not supplemented according to the comments of the reviewers, corresponding supplements and discussions were made in the text. I think it is acceptable for publication.